# Electrical Stimulation of Mesenchymal Stem Cells as a Tool for Proliferation and Differentiation in Cartilage Tissue Engineering: A Scaffold-Based Approach

**DOI:** 10.3390/bioengineering11060527

**Published:** 2024-05-22

**Authors:** Nicolas Lehmenkötter, Johannes Greven, Frank Hildebrand, Philipp Kobbe, Jörg Eschweiler

**Affiliations:** 1Department of Orthopaedic, Trauma and Reconstructive Surgery, RWTH Aachen University Clinic, Pauwelsstraße 30, 52074 Aachen, Germany; fhildebrand@ukaachen.de; 2Department of Thoracic Surgery, RWTH Aachen University Clinic, Pauwelsstraße 30, 52074 Aachen, Germany; jgreven@ukaachen.de; 3Department of Trauma and Reconstructive Surgery, BG Klinikum Bergmannstrost Halle, Merseburger Straße 165, 06112 Halle (Saale), Germany; philipp.kobbe@bergmannstrost.de (P.K.); joerg.eschweiler@bergmannstrost.de (J.E.); 4Department of Trauma and Reconstructive Surgery, University Hospital Halle, Ernst-Grube-Straße 40, 06120 Halle (Saale), Germany

**Keywords:** electrical stimulation, mesenchymal stem cells, osteoarthritis, scaffold, cartilage, chondrogenesis, chondrogenic differentiation, proliferation

## Abstract

Electrical stimulation (ES) is a widely discussed topic in the field of cartilage tissue engineering due to its ability to induce chondrogenic differentiation (CD) and proliferation. It shows promise as a potential therapy for osteoarthritis (OA). In this study, we stimulated mesenchymal stem cells (MSCs) incorporated into collagen hydrogel (CH) scaffolds, consisting of approximately 500,000 cells each, for 1 h per day using a 2.5 $V_{pp}$ (119 mV/mm) 8 Hz sinusoidal signal. We compared the cell count, morphology, and CD on days 4, 7, and 10. The results indicate proliferation, with an increase ranging from 1.86 to 9.5-fold, particularly on day 7. Additionally, signs of CD were observed. The stimulated cells had a higher volume, while the stimulated scaffolds showed shrinkage. In the ES groups, up-regulation of collagen type 2 and aggrecan was found. In contrast, SOX9 was up-regulated in the control group, and MMP13 showed a strong up-regulation, indicating cell stress. In addition to lower stress levels, the control groups also showed a more spheroidic shape. Overall, scaffold-based ES has the potential to achieve multiple outcomes. However, finding the appropriate stimulation pattern is crucial for achieving successful chondrogenesis.

## 1. Introduction

Osteoarthritis (OA) is one of the most common diseases among adults. A National Health Survey in the United States has shown a prevalence of 37.4% for adults, while 12.1% were symptomatic [1]. Radiographic changes in articular cartilage (AC) could be detected in the majority of people by age 65 [2]. This leads to a major financial burden, with direct costs ranging from USD 1442 to USD 21,335 and indirect costs from USD 238 to USD 29,935 per patient in the United States [3], costing the United States’ economy more than USD 60 billion per year [4]. AC is remarkably durable and functionally efficient under normal circumstances. It plays a crucial role in load distribution, peak stress mitigation, and maintaining low friction levels, all of which contribute to its long-lasting performance [5]. However, the onset of OA disrupts this equilibrium.

Although the pathogenesis of OA is not fully understood, it is believed that imbalances in AC homeostasis between anabolic and catabolic activity lead to irreversible degradation of the extracellular matrix (ECM) of AC [6,7,8]. These imbalances are created by multiple factors, including abnormal mechanical loading, aging, and genetic alterations [8,9]. Unlike bone, AC undergoes no internal remodeling and turnover throughout life [7,10]. Because of this and its hypovascularity, its capabilities of regeneration and repair are limited [6].

In general, cartilage can be categorized into three subgroups, hyaline cartilage, fibrocartilage, and elastic cartilage, differing in the composition of their ECM [11]. AC consists of hyaline cartilage, characterized by type II collagen (COL2) accounting for approximately 90–95% of the total collagen content [12]. This knowledge is important to understand the challenges of defect healing and current treatment options for OA. Treatment of OA consists of conservative and surgical options. Conservative approaches include hyaluronic acid, platelet-rich plasma [13], and physical therapy [14]. Surgical approaches are reparative methods like microfracturing, forming fibrous cartilage [15,16]; and also restorative methods like osteochondral autologous transplantation [17], autologous chondrocyte implantation [18]; or scaffold-based methods such as matrix-induced autologous chondrocyte implantation, producing physiological hyaline cartilage [15,19,20]. Total joint replacement surgeries in the form of endoprosthetics are used as a last resort [21,22]. An effective treatment option has not been discovered yet, which is why there is a demand for in vitro studies to discover new methods contributing to the healing of defective AC. The goal of cartilage tissue engineering is to overcome the limited regeneration capabilities and recreate physiological AC with the potential to substitute AC in a joint as a treatment for OA. To achieve this, hyaline cartilage has to receive signals promoting differentiation, proliferation, and viability. There are many approaches to this, including chemical [23] and biophysical stimuli such as electrical and mechanical stimulation [24,25].

One strategy is to use mesenchymal stem cells (MSCs), being progenitor cells that are capable of differentiating into chondrocytes [26], and attempt to induce chondrogenic differentiation (CD). MSCs offer the benefit of self-renewal and easy accessibility, and have the ability to differentiate into all three lineages (ectoderm, mesoderm, and endoderm) [27]. MSCs can be harvested from various sources including bone marrow [28,29], peripheral blood [30], adipose tissue [29], dental pulp [31], and menstrual blood [32].

Unfortunately, MSCs alone cannot achieve the desired outcome. As a result, researchers are increasingly investigating standardized methods, such as ES, that can be externally controlled to improve the outcome. Electric signals are a natural part of our physiology. However, utilizing them as a tool for tissue engineering is a relatively new field of research [33]. When stimulating MSCs for AC research, several positive effects could be observed. These include proliferation while maintaining multipotency [34], as well as research showing CD [35,36,37,38,39,40] and MSC reorganization [41]. This suggests that ES can effectively direct MSCs towards developing into chondrocytes, with this being the key to forming hyaline cartilage [42]. The effects of ES are influenced by numerous factors, including voltage, waveform, frequency, and duration of stimulation, among others. There are also different methods of stimulus delivery: direct, capacitive, inductive, or combined stimulation [43]. These factors offer almost infinite possibilities for parameter settings, necessitating extensive in vitro experimentation. Furthermore, this poses challenges in comparing results and interpreting them within the literature.

Throughout recent years, hydrogels have been utilized as scaffolds due to their viscoelastic nature and hydrated conditions that closely mimic those of native tissues, along with their significant equilibrium swelling capabilities facilitating the transfer of signaling molecules, nutrients, and metabolic wastes [44]. They allow tunability, can be 3D printed, can host different cell types for implantation into human cartilage [45], and have been shown to induce chondrogenic differentiation of MSCs after three weeks [46].

In this study, an electrical stimulation device was developed to deploy ES on MSCs by direct coupling. The device was designed to explore the potential of ES as a method to promote CD and enhance cell proliferation, while concurrently minimizing the stress experienced by MSCs. Additionally, the study aimed to establish a standardized experimental setup, which can enable systematic exploration of various frequencies and voltages to allow a reproducible examination of the effects of ES on MSCs.

## 2. Materials and Methods

### 2.1. Experiment Design

The experiment was designed to have 4 different time points at 24 h, 4 days (D4), 7 days (D7), and 10 days (D10). The 24h time point is only a control to have a baseline for comparison. For the D4, D7, and D10 time points, there was a control group as well as an ES group to observe changes caused by electrical treatment. Every group contained 6 scaffolds: 4 were used for gene analysis, 1 was used for two-photon imaging, and 1 was used for staining.

### 2.2. Cell Culture

Bone marrow-derived MSCs obtained from adult femur heads after total joint replacement were cultured and expanded at passage 2 in DMEM, low glucose, GlutaMAX™ medium (Thermo Fisher Scientific Inc., Darmstadt, Germany) with the addition of 10% fetal calf serum (PAN-Biotech GmbH, Aidenbach, Germany), 10% Pen/Strep (Sigma-Aldrich^®^, Merck KGaA, Darmstadt, Germany), and 10% NEAA (Gibco, Thermo Fisher Scientific Inc., Darmstadt, Germany) for 4 days in T175 flasks in the incubator at 37 °C and 5% CO_2_.

### 2.3. Collagen Type I Hydrogel Scaffolds

The scaffolds were produced using a collagen type I hydrogel (CH) (Collagen-NF 10 mg/nl-©meidrix biomedicals GmbH, Esslingen am Neckar, Germany). The product contains two components: the CH and an activation solution (AL) to trigger pH-activated gelation when mixed in a 4:1 ratio. We used autoclaved silicon molds measuring 8 mm in diameter and 6 mm in height, creating scaffolds with a volume of approximately 300 mm^3^. To seed the cells into the scaffolds, the first step was to harvest them from the T175 flasks. The cells were washed two times with PBS, and then, enzymatically dissociated with trypsin. The cells were then washed with 12 mL of medium, counted, and centrifuged at 1200 rpm for 10 min to obtain a cell pellet. After carefully aspirating the supernatant, the cell pellet was mixed with the AL to a final concentration of approximately 500,000 cells/60 μL AL. The molds were then prefilled by pipetting in 240 μL of CH. In a second step, 60 μL of the AL–cell mixture was pipetted to each chamber and mixed vigorously to create a homogeneous scaffold with a volume of 300 μL, containing approximately 500,000 cells. To complete the gelation, the molds were incubated at 37 °C for 20 min. After that, the scaffolds were removed from the mold by injecting PBS with a syringe and a thin cannula from the side to release them without damage, and were then incubated in 6-well plates separately in the medium at 37 °C overnight.

### 2.4. Stimulation Chamber

Custom ES devices were used (Figure 1). These were adapted to a 6-well plate by replacing the lid to complete the ES chamber. They hold two L-shaped platinum electrodes 15 mm in width and separated by 21 mm for each well. They were immersed into the wells to generate an electric field and current flow in the medium in the form of direct coupling. The electrodes have been measured to be submerged 3 mm below the top of the scaffold to provide a uniform electric field throughout the scaffold. The platinum electrodes have been tested to be non-toxic [47], non-reactive, and biocompatible [48,49]. The scaffolds were positioned in the center of a well by using autoclavable insert discs equipped with an inner ring capable of supporting the scaffolds. Due to the tendency of the scaffolds to float, caused by air inclusions, they were fixed to the insert discs using TISSEEL 2 mL fibrin adhesive (Baxter Deutschland GmbH, Unterschleißheim, Germany). Using this setup, the scaffolds had no direct contact with the electrodes, minimizing cell toxicity. Before stimulation, the devices were autoclaved daily and the actual stimulation was carried out inside the incubator. Once it was completed, the stimulation devices were replaced by the 6-well lids under a laminar flow hood to minimize any risk of contamination. Then, they were autoclaved before the next use.

### 2.5. Signal Generator

An additional device for the electrical modulation was developed and built up to produce a sinusoidal voltage signal. It makes use of an Arduino Uno to control a waveform generator on a separate circuit board. The free-programmable AD9833 waveform generator (Analog Devices Inc., Wilmington, MA, USA), capable of producing sine, rectangular, and triangular outputs at a frequency range of 0 to 12.5 MHz, is the main part of the PCB. It can be programmed using the Arduino Uno over a three-wire SPI interface giving different opportunities for signal programming. Programming of the Arduino Uno was conducted in the Arduino IDE (Version 2.0.0-rc6). The produced signal is amplified by a non-inverting operational amplifier circuit using the LM358 operational amplifier (Texas Instruments Inc., Dallas, TX, USA) capable of amplifying the signal to approximately 20 $V_{pp}$ and could be adjusted by a potentiometer. The signal is filtered through a 100 μF capacitor to remove the DC component and create an alternating signal around the 0 V baseline.

### 2.6. Electrical Stimulation

Before this experiment, non-published testing of voltage and frequency was performed using two-photon imaging with an Invitrogen (Thermo Fisher Scientific Inc., Darmstadt, Germany) LIVE/DEAD^®^ Cell Imaging Kit and hematoxylin and eosin (HE) staining to assess cell viability, showing that increasing the frequency of the sine wave drastically increased cell death at a voltage of 2 $V_{pp}$. Using low frequencies up to 10 Hz and high voltages up to 10 V did not seem to trigger cell death. However, for voltages above 2.5 V electrolysis of the medium was observed.

Inactivation times of voltage-gated calcium channels (VGCCs), identified as a crucial element of the chondrogenesis mechanism with ES [39], range from 20 to 110 ms [50]. Considering the experimental findings and physiological fundamentals, the ES was set to a 2.5 $V_{pp}$ (119 mV/mm) sine wave at a frequency of 8 Hz for 1 h/d (Figure 2). The voltage adjustment was conducted using a potentiometer via simultaneous analysis of the signal with a digital oscilloscope (GWinstek GDS-1052-U—GOOD WILL INSTRUMENT EURO B.V., Veldhoven, The Netherlands).

### 2.7. Gene Analysis

The RNA was extracted using the Omega Biotek RNA-Solv^®^ (VWR International GmbH, Darmstadt, Germany) Reagent before real-time qPCR. The scaffolds were mechanically released from the insert disks and washed with PBS. As previous RNA extractions have shown that the CH scaffolds are digested in the first RNA extraction step, no additional digestion was necessary. The RNA extraction was performed following the manufacturer’s protocol. The RNA concentrations and purity values were determined using a NanoDrop Spectrophotometer (Thermo Fisher Scientific Inc., Darmstadt, Germany). The RNA was converted to cDNA using the High-Capacity RNA-to-cDNA™ Kit by Applied Biosystems™ (Thermo Fisher Scientific Inc., Darmstadt, Germany). The primers were designed with the help of the NCBI gene database and manufactured by Eurofins Germany (Eurofins NDSC Food Testing Germany GmbH, Hamburg, Germany) and each was validated by a 10-fold serial dilution series of six dilutions to be in the range of 93–107% efficiency (Table 1). For reliable PCR results, triplets of four biological groups for each experiment group were used. The gene expression levels were normalized to glycerinaldehyde-3-phosphate dehydrogenase (GAPDH), β2-microglobulin (B2M), and peptidylprolyl isomerase A (PPIA) housekeeping genes. The analysis of aggrecan (ACAN), collagen type I (COL1), collagen type II (COL2), SRY-box transcription factor 9 (SOX9), and matrix-metallo-protease 13 (MMP13) was performed using the RT-qPCR CFX Opus 96 from Bio-Rad (Hercules, CA, USA). Ct values were calculated using the delta-delta Ct method in a Microsoft Excel (Microsoft Corporation, Redmond, WA, USA) spreadsheet.

### 2.8. Two-Photon Imaging

Using two-photon imaging, viability, proliferation, and cell morphology, consisting of the sphericity and volume of the MSCs inside the scaffold, could be assessed. For this purpose, CH scaffolds were stained as a whole using the Invitrogen LIVE/DEAD^®^ Cell Imaging Kit (Thermo Fisher Scientific Inc., Darmstadt, Germany). To ensure reproducibility, 500 μm stacks were imaged with a z-step size of 2 μm from three areas of each scaffold with the Leica Stellaris 8 microscope (Leica Camera AG, Wetzlar, Germany). The kit stains live cells green and nucleic acids red, which allows for measurement of the dead component in red particles rather than cells. While this method cannot provide an exact measurement of dead cells, it can reveal trends. The analysis of two-photon images and data extraction for statistical analysis was performed using the Imaris Microscopy Image Analysis Software (Version 10.0). The measured variables were live cell count, live/dead ratio, volume, and sphericity.

### 2.9. Scaffold Measurements

During the experiment, the scaffolds were measured manually, using a digital caliper with a resolution of 0.01 mm, at D7 and D10. Another experiment was performed where the potential effect of CD markers on scaffold shrinkage was examined. For this purpose, both MSCs and CHOs were embedded in CH scaffolds with approximately 500,000 cells each and incubated for 14 days with or without CD medium, which included the above-mentioned composition for MSCs and High-Glucose DMEM Medium (Thermo Fisher Scientific Inc., Darmstadt, Germany), including Pen/Strep (Sigma-Aldrich^®^, Merck KGaA, Darmstadt, Germany), fetal calf serum (PAN-Biotech GmbH, Aidenbach, Germany), and ITS Plus (1:100, Gibco, Thermo Fisher Scientific Inc., Darmstadt, Germany), and for the designated groups the CD factors (CDFs) transforming growth factor ß1 (0.01 μg/mL), dexamethasone (100 nM), sodium pyruvate (1 mM), ascorbic acid (0.17 mM), and proline (0.35 mM). Each group contained five scaffolds, except the CHO group without CDF. This group contained only four scaffolds. A medium change was performed every three days. After 14 days they were also measured via a digital caliper and weighed with a precision scale (KERN & SOHN GmbH, Balingen-Frommern, Germany—precision 0.0001 g).

### 2.10. Histological Analysis

Scaffolds were fixed in 4% paraformaldehyde overnight. The samples were then dehydrated and embedded in paraffin. Sections were cut using a microtome to produce 2 μM slices. Samples were stained with HE and COL2 stainings using anti-collagen type II (rabbit) antibody (Rockland Immunochemicals Inc., Limerick, PA, USA) at a 1:100 dilution according to internal protocols.

### 2.11. Statistical Analysis

Statistical analysis of cell viability and proliferation as well as scaffold weight and shrinkage was performed using a two-tailed unpaired Student’s *t*-test as a powerful test for small sample sizes. Using the Shapiro–Wilk test, data normality was tested and it was found that the volume and sphericity were not normally distributed. Therefore, Kruskal–Wallis tests were performed. To determine the significance of the PCR results, one-way ANOVA and Tukey–Kramer tests were used. All statistical analyses were carried out in the R programming language (version 2023.09.1+494) at a significance level of 5%. *p*-values of the different tests are reported throughout.

## 3. Results

### 3.1. Morphology, Proliferation, and Viability

#### 3.1.1. Scaffold and Cell Morphologies

During the experiment, a visual reduction in the volume of the stimulated scaffolds was observed (Table 2). At D7, there was a 34.9% reduction in volume between the control and treated groups. At D10, the effect was even greater at 66.4%, while the control groups maintained the same volume (Figure 3g). This indicates a significant difference between the control and stimulated scaffolds at D7 (*p* < 0.01) and D10 (*p* < 0.001). The effect of scaffold shrinkage is also greater at D10 than at D7 (*p* < 0.05), suggesting a relationship between stimulated days and scaffold volume.

The stainings in Figure 3a–d allow for a visualization of the scaffold structure. Especially in Figure 3a,b, an overview of the scaffolds can be seen. The texture of the hydrogel can be observed with its internal pathways and incorporated air bubbles that are ideal for cell colonization. Figure 3c shows a close-up of the stimulated cells in HE staining. The classic spindle-like morphology can be seen. Figure 3d shows the COL2 expression of stimulated cells. In the 3D reconstruction of two-photon image stacks (Figure 3e,f), the classic spindle-like morphology of MSCs can also be observed.

The additional experiment in Figure 4 shows that adding CDF to the medium either for MSCs or CHO significantly reduces their volume and weight after 14 days (*p* < 0.001). Scaffolds containing MSCs and incubated with CDF exhibited a 6.48-fold decrease in volume and a 6.23-fold decrease in weight compared to scaffolds incubated without CDF. A similar effect was observed for CHOs, with scaffolds incubated without CDF exhibiting an 8.49-fold decrease in volume and an 8.51-fold decrease in weight compared to controls without CDF.

#### 3.1.2. Growth and Viability

Analysis of the two-photon image stacks shows a reduction in live cells after the 24 h control to the D4 control, although not significant (*p* = 0.071) (Figure 5a).

After the 24 h control, the ES group regenerates and proliferates faster than the control group and reaches its peak at D7 before a reduction in live cells occurs at D10. At D4, there is a 2.1-fold increase in live cells in the ES group (not significant, *p* = 0.059), at D7 2.7-fold (*p* < 0.001) (Figure 3e,f), and at D10 1.86-fold (*p* < 0.05). There is also a higher amount of dead cells at D4 (not significant, *p* = 0.139) and D7 (not significant, *p* = 0.062) in ES groups compared to controls, although not significant. Except for D4, the live/dead ratio is also higher for the ES group (Table 3).

Due to the method of consistently comparing the same dimensions of scaffold areas, the shrinkage of the scaffolds was not considered. As a result of the shrinkage of the stimulated scaffolds (Table 2), there must have been a greater number of cells in the same area when compared to the controls. To quantify the proliferation of MSCs in the ES groups, without the shrinkage effect, we took into account the RNA yield (Figure 5b), which provides a more reliable measure. The mean RNA concentrations of the ES groups at D4 and D7 were significantly higher than those of the controls, with a 5.4-fold increase at D4 (*p* < 0.05) and a 9.5-fold increase at D7 (*p* < 0.05). After 10 days, the mean RNA concentrations of the control group increased to just below the stimulated group. The amount of RNA varied in this group, resulting in a higher standard deviation, and thus, no significant difference to the ES group.

#### 3.1.3. Cell Volume and Sphericity

The 24 h control group exhibited the highest values for both volume and sphericity metrics. In comparison, the stimulated groups consistently demonstrated higher volumes relative to the control groups, while the control groups’ volumes remained relatively stable over time. The data presented in Figure 6a suggest that ES induces MSC hypertrophy, with D4 and D7 showing a statistical significant difference, while D10 shows a difference, but not significant (*p* = 0.131).

From D4, with a sphericity of 0.744 in the control group and 0.728 in the ES group, there is a noticeable and consecutive reduction in sphericity through D7 (control: 0.7331, stimulated: 0.663) continuing to D10 (control: 0.651, stimulated: 0.624). This trend signifies a loss of 12.51% in sphericity within the control group and a 17.01% reduction in the ES group when comparing the measurements on D10 to those of the initial 24 h control. It is also noted that the ES groups consistently exhibit reduced sphericity in comparison to the control groups, with D4 showing a decrease of −2.07% (not significant, *p* = 0.113), D7 presenting a −9.27% reduction (*p* < 0.001), and D10 showing a −4.14% decrease (*p* < 0.05) (Figure 6b).

### 3.2. Gene Expression Analysis

Figure 7 illustrates the results of RT-qPCR analysis relative to the 24 h control group. The expression of ACAN gradually increases from D4 to D10 in the stimulation groups, exhibiting a 2.6 to 4.3-fold increase compared to the 24 h group, with a significant difference between the ES and control groups at D4 (*p* < 0.001) and D7 (*p* < 0.001). Although it remains constant at D4 and D7, the expression of ACAN in the controls rises at D10 to the same level as the ES group.

The expression of COL1 remains consistently lower than the 24 h baseline throughout all time points. In the ES groups, there is a gradual decrease over the time points in COL1 expression. However, in the control groups, COL1 expression fluctuates. No significant differences between the groups are found. Although the expression of COL2 is below the 24 h control for both the ES and control groups on D4 and D7, a noticeable increase in COL2 expression is detected on D10. The ES groups exhibit an 11.9-fold increase in expression, while the controls exhibit an 8.8-fold increase in expression (D7C–D10C: *p* < 0.001; D7S–D10S: *p* < 0.001). On D10 there is also a difference between the ES and control groups (*p* < 0.05). The increase in COL2 expression of the control groups observed on D10 is similar to that of ACAN. In contrast, SOX9 expression is significantly lower at D10 compared to at 24 h, with all groups showing a greater than 0.65-fold decrease in expression. Similar to COL1, the ES groups exhibit a decreasing trend, while the controls fluctuate. At D4 (*p* < 0.001) and D7 (*p* < 0.05), there is a significant reduction in SOX9 expression in the ES groups. Additionally, while the control groups maintain a consistently low level of MMP13 expression, the expression gradually increases from D4 to D10 in the ES groups, with a significant up-regulation on D10 compared to the control (*p* < 0.001), D7 ES group (*p* < 0.01), and D4 ES group (*p* < 0.05).

## 4. Discussion

The purpose of this study was to investigate the impact of ES on MSC-seeded collagen scaffolds, to contribute to the mixed results in the existing literature on ES, and to positively influence the research on AC substitution. Many studies about ES in AC tissue engineering have focused on the effects of ES in AC in vivo or on chondrocytes [24,33,43,51,52,53], while only a limited number of studies have examined MSCs and their differentiation potential, particularly in the direction of hyaline cartilage [34,35,36,37,38,39,40,41,54]. These studies have produced mixed results, including strong [39] and weak [38] CD, proliferation [34], reorientation [41], and no differentiation with full capacity to differentiate into all lineages [34]. Due to the wide variety of settings that can be applied to ES, it is difficult to establish comparability because, in theory, only a small change in a single setting could alter the way cells respond to ES. The types of ES employed varied: direct current and alternating signals with different waveforms such as sinusoidal and cubic were used. The voltage ranges used in prior studies were between 10 mv/cm and 25 V/cm. The frequencies used ranged between 5 Hz and 448 KHz. Also, there are many ways of timing the ES. For example, every x hours for a specific duration of stimulation with breaks [34] or without breaks [37]. Another distinction must be made in the application of ES between a direct, a capacitive, an inductive, and a combined coupling approach [43].

The mechanisms underlying the effects of ES on MSCs are poorly understood and require further exploration. Studies have identified several channels and receptors, including VGCCs, transient receptor potential channels, and purinergic receptors, particularly P2X4, mediating Ca^2+^ and ATP oscillations that appear to influence the effects of ES on MSCs [39,55,56]. These findings are based on the assumption that VGCCs are not only present in excitable cells but also in other cells such as MSCs. The influence of VGCCs in chondrogenesis can be confirmed by the discovery that calcium inhibitors impede chondrogenesis, as well as the proliferation and hypertrophy of chondrocytes [57]. Also transforming growth factor-β (TGF-β) and bone morphogenetic protein (BMP) signaling seems to be important during CD, with TGF-β also inducing pre-CD [58].

As another hypothesis, it has been proposed that ES may temporarily increase the permeability of the plasma membrane, thereby inducing a form of cell stress through electrical stress [55], which is also shown by the up-regulation of MMP13 (Figure 7) in our study, and transmembrane ion transport and a burst of intracellular Ca^2+^ [59]. All in all, the Ca^2+^ signaling and subsequent cAMP/PKA signaling seems to be a central aspect in the mechanism of the induction of chondrogenesis by ES [60].

In this study, we examined the proliferation behavior and stress on cells by two-photon imaging with live/dead fluorescence and by taking into account the RNA yield. The live/dead ratio was low in the 24 h control group due to cell stress during the production of cell-seeded collagen scaffolds. After the initial stress, the ES groups showed a higher mean cell number at D4, although not significant, and a significantly higher mean RNA concentration. At D7, the most pronounced effect was observed, with a 2.7-fold higher mean cell number and a 9.5-fold higher mean RNA concentration compared to the controls. D7 seems to be the sweet spot for increased proliferation. The effect seems to diminish at time point D10. Vaca-Gonzalez et al. already reported this, claiming MSCs only proliferate until day 7 [35]. While there is a significantly higher cell number, explained by the shrinkage of the scaffold at D10, the difference in mean RNA concentration is not significant due to the increase in RNA concentration in the control groups, which may be an effect of the CH scaffold that can also be seen at D10 for ACAN and COL2 values. The change in absolute cell number must be seen as a combination of proliferation and shrinkage of the simulated scaffolds. RNA yields show a more meaningful value that is not related to shrinkage but could be affected by higher transcription due to ES. Based on this analysis, it was found that ES induced proliferation, with a significant difference compared to the control groups, especially on D7. However, it was also observed that ES induces cell stress, resulting in a higher number of dead particles in the ES groups. These results show some similarities, but also differences to what has already been found in previous studies [34,35,36,37,38,39,40,41,54] and mixed results in general. Hernandez-Bule et al. [34] also reported that ES enhanced the proliferation of MSCs but only achieved a 20% increase in cell number when applied to cells at low passages using 448 KHz short pulses with direct coupling. Mardani et al. and Esfandiari et al. also found an ES-induced proliferation, but it was not significant [38,40]. This effect is one of the most prominent findings of this experiment and can already be considered as a success of cells reacting to ES.

It is unclear if CD was achieved due to the various factors both supporting (cell volume, scaffold shrinkage, ACAN and COL2 expression) and disagreeing (cell sphericity, SOX9 expression) with CD. Current studies on ES on MSCs have not examined the cell volume of the stimulated cells [34,35,36,37,38,39,40,41,54]. Therefore, a comparison with these results is not possible. The values of mean volume and sphericity of the 24 h control group are affected by an incomplete adhesion of the MSCs to the CH scaffold, making them more round and voluminous, hence this group displays the highest volume and sphericity. At D4, this process is fully completed. Studies that examined CD have generally shown that MSCs are hypertrophic during this process. Hypertrophy occurs during late CD and enchondral ossification of chondrocytes and also occurs when TGF-β is used to chemically induce CD but it leads to the creation of lower-quality cartilage tissue [23,61]; so, the significantly higher volume of stimulated cells at all stimulated time points observed in this experiment can be seen as an indication of CD. However, it might also be an effect of higher cell stress leading to necrosis and further swelling of MSCs [62]. As a result, the aspect of volume should be further investigated in future research on ES and CD.

In contrast to cell volume, sphericity has been examined as a relevant factor for CD by Vaca-Gonzalez et al. [35]. They found that stimulated cells had a more circular shape, indicating CD, as it is generally accepted that cells in a 3D environment that have a spherical shape are indicative of CD [63]. The results in this experiment imply the opposite: The D7 and D10 control cells displayed significantly higher sphericity, indicating that ES adversely affected cell roundness, and therefore, could be a possible indication against chondrogenesis. Since there is only one publication that also examines sphericity, it is difficult to place our results in the context of the literature. Similar to cell volume, this should be further investigated in the future.

Even though not expected, a shrinkage of the scaffolds occurred and was visible from D7. Following this, the scaffolds were measured at D7 and D10 and a significant volume change was found for the stimulated scaffolds. This phenomenon, referred to as bio-actuation, is a well-known effect that typically occurs when CD, particularly TGF-β, is added to the cell medium. This has also been observed and confirmed in previous internal experiments, that show not only a significant volume reduction (*p* < 0.001) of the scaffolds but also a significantly lower weight (*p* < 0.001), which could be the result of a loss in water accompanying bio-actuation (Figure 4). It is based on the mechanism of cell contraction through integrin-mediated collagen–cell contact during chondrogenesis [64]. Being a TGF-β-mediated effect, this suggests once again that the ES applied in this experiment induced chondrogenesis.

Consistent with this, Kwon et al. [39] found that ES of MSCs in a 3D-micromass culture triggers an increase in TGF-β1 and BMP2 expression, that induce CD, when applying high voltages up to 25 V/cm at 5 Hz, 8 ms for 3 days with direct coupling. However, only TGF-β causes pre-chondrogenic condensation [58], which was also found. The study also found strongly increased expression of COL2 (66-fold), ACAN (43-fold), and SOX9 (35-fold), and decreased expression of COL1 compared to the control, strongly indicating chondrogenesis. The differences in the expression of chondrogenic markers are much higher than we found, which might suggest an advantage of this cultivation system. We found an up-regulation of ACAN on D4 (2.6-fold) and D7 (4.3-fold) but not at D10. Here, the control groups caught up to the stimulation group. This behavior reminds one of the pattern of proliferation and might be explained by the duration required for MSCs to undergo CD [65]. Moreover, since the scaffold may also prompt CD, the control groups exhibited a delayed rise. For COL2, the only significant difference can be found at D10. Here, the control group and the stimulation group increase expression. This is similar to the pattern of ACAN expression and can be attributed to the same effect mentioned before. The increased expression of COL2 in the ES groups on D10 and ACAN on D4 and D7 underlines that ES has a positive potential for chondrogenesis. Kwon et al. [39] did not provide any evidence of cell stress. Especially after 10 days, a significant expression of MMP13 (over 100-fold) could be found in our study, and at all time points MMP13 levels were higher in the ES groups than in the controls. Given our findings and the fact that voltages above 2.5 V lead to electrolysis of the medium and high expression of MMP13, we speculate that the ES by Kwon et al. [39] must have induced significant cell stress, which could explain the short stimulation time of only three days. Similar, but not as high, results of increased CD were found by Vaca-Gonzalez et al. [35], Zhang et al. [36], Hiemer et al. [37], and Esfandiari et al. [40]. Comparable to our results, Mardani et al. [38] found increased, but partially not significant expression of CD markers. As there was significant up-regulation of COL2 and ACAN but also significant down-regulation of SOX9 in our study, this does not fully imply CD, even though the pro-chondrogenic factors outweigh the anti-chondrogenic factors. This can be explained by the short stimulation time in our study, as Vaca-Gonzalez et al. found an up-regulation of SOX9 beginning on day 14 [24].

Overall, our study revealed increased cell numbers and partial up-regulation of pro-chondrogenic genes (COL2, ACAN), but also down-regulation of SOX9. We observed lower cell sphericity but a higher volume of ES group cells, and a shrinkage of scaffolds in the ES groups. Our findings indicate clear effects of ES on MSCs, although its full support of CD is lacking. Since CD is a time-consuming process that has been shown to reach its full extent after 14 to 35 days [24,65,66], it is recommended to extend the duration of ES to fully investigate its effect in the full process of differentiation. It is important to note that ES causes a high level of cell stress, as shown in Figure 7e, which could be problematic when increasing the duration of ES and is a weakness of this stimulation method. In further studies, proliferation markers such as Ki-67 should be used to exclude the effect of scaffold shrinkage. We believe that ES has the potential to induce CD, as well as differentiation into other lineages. Because of this, other markers should be introduced to fully understand the effect the stimulation has on the MSC. Although these scaffolds display similar biomechanical properties to native AC, they do not resemble the structure of AC with its multiple zones and collagenous arcuate architecture [5], making AC so unique and difficult to recreate. In future experiments, it is desirable to adjust the electrodes to create a more homogeneous electric field due to their L-shaped form. Our test setup, incorporating a microcomputer in the form of an Arduino with its free programmability, makes the possibilities of ES patterns limitless, making it the ideal tool for the further research that needs to be conducted to further explore the field of ES.

## 5. Conclusions

In this study, we have explored the effects of ES on MSCs within CH scaffolds, a promising avenue in cartilage tissue engineering. Our findings have shown that ES indeed has the potential to enhance both the proliferation and CD of MSCs. This research contributes to the understanding of the effects of ES in cartilage tissue engineering. It highlights the potential benefits of this approach, but also underscores the necessity for more research on alternating stimulation protocols. It leaves many questions regarding the effect of ES on MSCs unanswered, as it is one of the first to address this research field and can be seen as a proof-of-concept and pilot study in the large field of opportunities ES provides. If the appropriate settings can be identified from the numerous configuration possibilities, ES has the potential to be a powerful tool in many applications, together with scaffolds, such as the CH scaffolds used in this experiment, that provide a suitable environment for MSCs that can be precisely matched to a patient’s cartilage defect, allowing patient-specific treatment. As research on ES continues, this study provides valuable insights that could lead to improved treatments for OA. Furthermore, if differentiation of MSCs into chondrocytes, forming hyaline cartilage, can be achieved, long-term studies of stimulated cells have to be conducted in vitro and in vivo to evaluate whether a treatment superior to current options can be established.

## Figures and Tables

**Figure 1 bioengineering-11-00527-f001:**
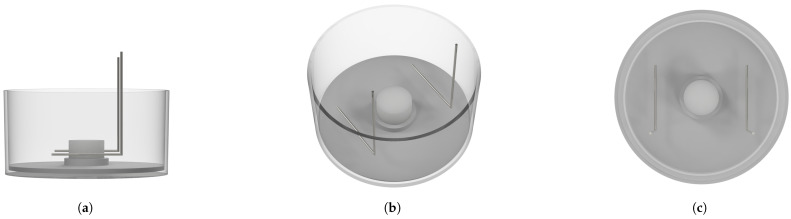
Schematic of a well of a 6-well plate containing the insert disks, scaffold, and two L-shaped platinum electrodes. (**a**) Shows the side view, (**b**) is an angled view, and (**c**) is the top view.

**Figure 2 bioengineering-11-00527-f002:**
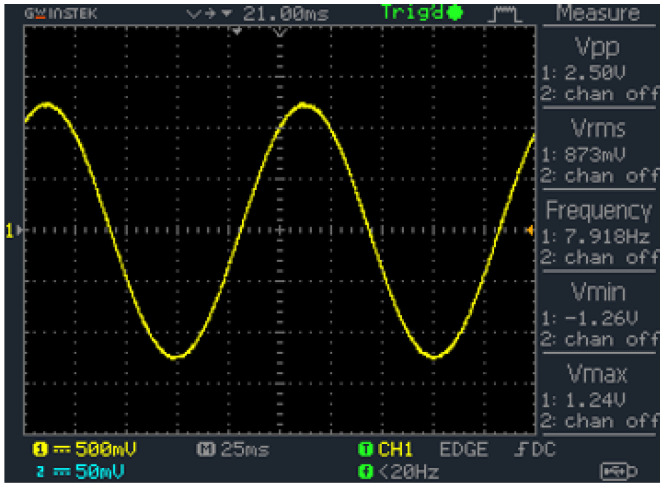
Screenshot of the oscilloscope showing the sine wave with 2.5 $V_{pp}$ and a frequency of ∼8 Hz.

**Figure 3 bioengineering-11-00527-f003:**
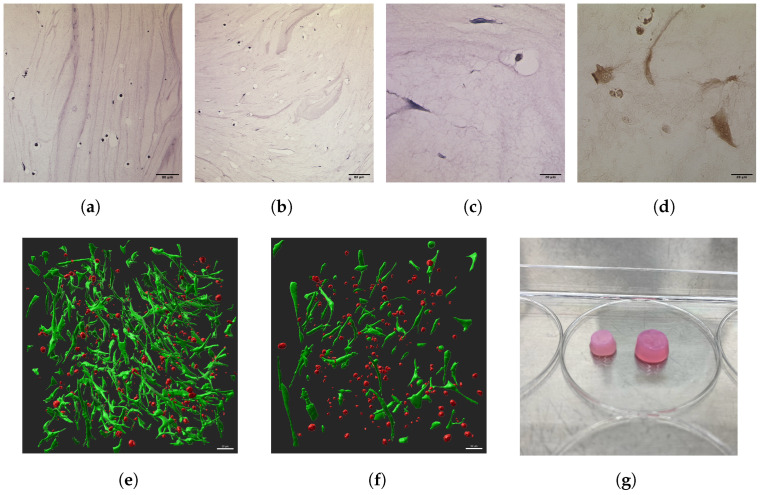
(**a**–**d**) Show photos of histological sections. (**a**,**b**) HE stainings of D4 at 10×: (**a**) is the control, (**b**) is the ES group. (**c**) An HE staining of the D7 ES group at 40×; (**d**) a COL2 immunohistochemical staining from the D10 ES group at 40× magnification. (**e**,**f**) Show 3D renderings of stacks from two-photon imaging from D7. (**e**) Displays the ES group; (**f**) the control group. (**g**) Shows a physical image of the scaffolds at D10. The left scaffold is from the ES group, the right scaffold is from the control group.

**Figure 4 bioengineering-11-00527-f004:**
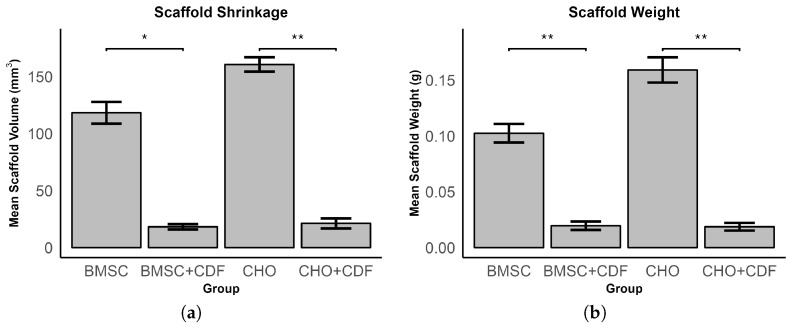
Comparison of the mean volume of scaffolds for different groups (BMSC: bone marrow-derived stem cells; CHO: chondrocytes; CDF: chondrogenic differentiation factors). (**a**) Displays the scaffold shrinkage, (**b**) displays the scaffold weight. Bars represent the mean volume and error bars show the standard error of the mean. Significance levels are indicated by asterisks (* *p* < 0.05; ** *p* < 0.01).

**Figure 5 bioengineering-11-00527-f005:**
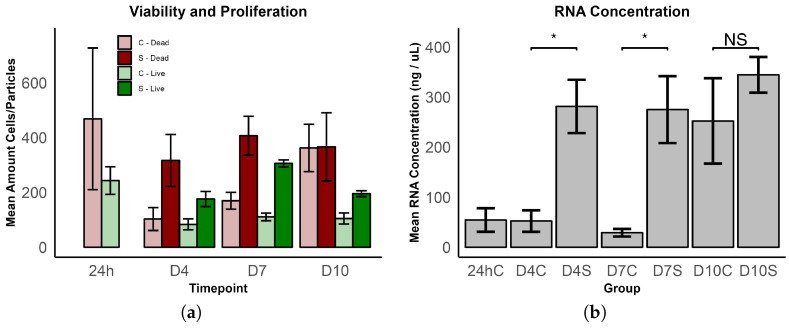
(**a**) Comparison of mean live cells/dead particles at different time points (24 h, D4, D7, D10) and conditions (control—C, stimulated—S). Bars represent the mean amount of counted cells for live cells or dead particles in two-photon image stacks. (**b**) Comparison of mean RNA concentrations at different time points (24 h, D4, D7, D10) and conditions (control—C, stimulated—S). Bars represent the mean RNA concentration. Significance levels are indicated by asterisks (* *p* < 0.05; NS: not significant).

**Figure 6 bioengineering-11-00527-f006:**
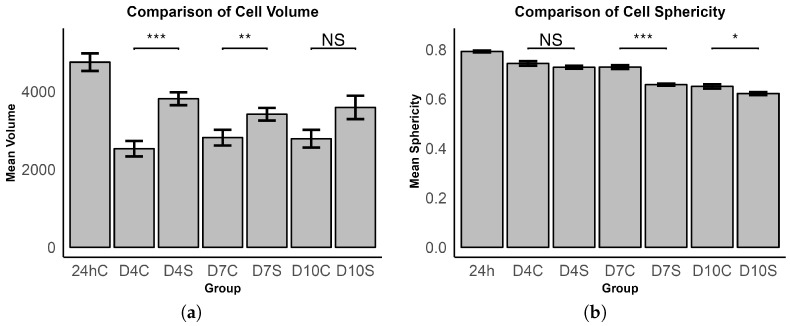
Comparison of (**a**) cell volume and (**b**) cell sphericity at different time points (24 h, D4, D7, D10) and conditions (control—C, stimulated—S). Bars represent the mean volume and error bars show the standard error of the mean. Significance levels are indicated by asterisks (* *p* < 0.05; ** *p* < 0.01; *** *p* < 0.001) and NS: not significant.

**Figure 7 bioengineering-11-00527-f007:**
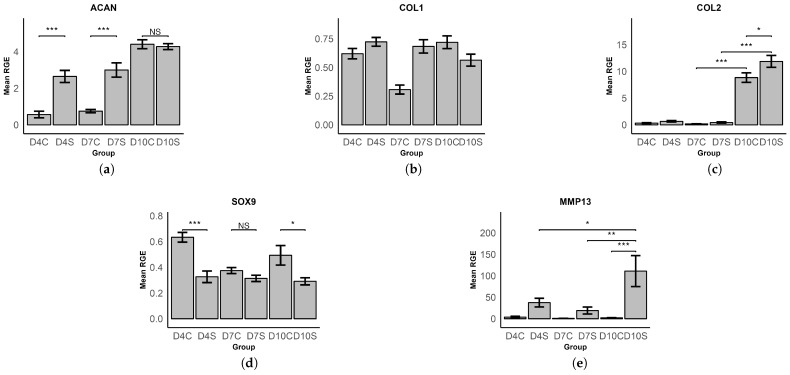
Mean relative gene expression (RGE) of (**a**) ACAN, (**b**) COL1, (**c**) COL2, (**d**) SOX9, and (**e**) MMP13 across the different time points (D4, D7, D10) and conditions (control—C, stimulated—S). Bars represent the mean RGE and error bars show the standard error of the mean. Significance levels are indicated by asterisks (* *p* < 0.05; ** *p* < 0.01; *** *p* < 0.001).

**Table 1 bioengineering-11-00527-t001:** Forward and reverse sequences of primers used for gene analysis.

Gene	Forward	Reverse
*GAPDH*	ACAACTTTGGTATCGTGGAAGG	GCCATCACGCCACAGTTTC
*B2M*	TGAGTATGCCTGCCGTGTGA	GCGGCATCTTCAAACCTCCAT
*PPIA*	CTTGGGCCGCGTCTCCTTT	TCCTTTCTCTCCAGTGCTCAGA
*ACAN*	TGCTATGGAGACAAGGATGAG	GATGAGGGGTCGGGGTA
*COL1*	TCTAGACATGTTCAGCTTTGTGGAC	TCTGTACGCAGGTGATTGGTG
*COL2*	TCCTCTGCGACGACATAATC	CAGTGGCGAGGTCAGTT
*SOX9*	AAGACGCTGGGCAAGCTCTG	GTAATCCGGGTGGTCCTTCTTG
*MMP13*	ATACTACCATCCTACAAATCTCGC	GCCAGTCACCTCTAAGCCG

**Table 2 bioengineering-11-00527-t002:** Mean volume of scaffolds in mm^3^.

Timepoint	Control	Stimulated
Day 7	175.9	114.6
Day 10	177.1	59.5

**Table 3 bioengineering-11-00527-t003:** The average counts of live cells, dead particles, and the calculated ratio across different experimental time points (24 h, 4 days, 7 days, and 10 days) and conditions (control—C, stimulated—S) are shown. Each cell represents the average count for the respective categories within the specific experimental group.

Group	Avg. Living Cells	Avg. Dead Particles	Ratio
24 h	243.7	468.7	0.520
D4C	83.7	103.0	0.812
D4S	176.0	317.0	0.555
D7C	110.7	169.7	0.652
D7S	306.0	407.0	0.752
D10C	105.0	362.3	0.290
D10S	195.7	366.7	0.534

## Data Availability

The data obtained in this study is available on request by the corresponding author.

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
