# Peer review of "Electrical Stimulation of Mesenchymal Stem Cells as a Tool for Proliferation and Differentiation in Cartilage Tissue Engineering: A Scaffold-Based Approach"

_bioengineering, 2024, doi:10.3390/bioengineering11060527_

Round 1

Reviewer 1 Report

Comments and Suggestions for Authors

Firstly, I would like to extend my heartfelt gratitude to all the esteemed academics who contributed to this valuable study. Given that the research was conducted on mesenchymal stem cells (MSCs), perhaps a few sentences could be added in the introduction regarding the differentiation capacities of MSCs and particularly their ability to be directed towards chondrogenic differentiation through electrical stimulation.

I would also like to make a suggestion for the discussion section. When it comes to the clinical application of MSCs, determining the optimal number of MSCs initially isolated from bone marrow or other sources becomes crucial. Considering the impact of voltage and frequency used for electrical stimulation on cell viability, advantages and disadvantages regarding the number of MSCs to be isolated can be compared with other studies.

Comments on the Quality of English Language

Overall, the quality of English is quite good. A few minor corrections may be needed. For example, in the 6th line, "per day" should be "per days"... In the 28th line, "lead to" should be "leads to"... In the 43rd line, "endoprosthetics" should be "endoprosthetic"... In the 101st line, "In a second" should be "In the second"... In the 191st line, "pyruvat" should be "pyruvate"... In the 202nd line, "wheight" should be "weight"... In the 386th line, "increases" should be "increase".

Reviewer 2 Report

Comments and Suggestions for Authors

In this study, the author stimulated mesenchymal stem cells (MSCs) incorporated into collagen hydrogel scaffolds for 1 hour per day using a 2.5 Vpp (119 mV/mm) 8 Hz sinusoidal signal, and found that scaffold-based ES had the potential to achieve multiple outcomes. However, the research lacked sufficient data and did not include in vivo experimental results. We can hardly judge the effectiveness of this method, as in vitro experiments are subject to many uncertainties and errors. Therefore, it is recommended to supplement with in vivo experiments and resubmit for publication. Meanwhile, there are some questions to be answered and some mistakes with formatting to be corrected in this manuscript.

1. Fig 1: Are there physical images of the scaffolds? Could you provide photos of the changes of the scaffolds after the stimulation? This would make it more visually informative.

2. Materials and Methods: the PCR experiment did not provide the sequence of the detected gene.

3. Fig 5: figure (a) did not have error bars. Did it only compare one set of data? It is recommended to add at least three sets of data and include error bars.

4. References: many of the references are quite old, and it is hoped that more recent articles from the past two years can be found to confirm its novelty.

Reviewer 3 Report

Comments and Suggestions for Authors

Manuscript “Electrical Stimulation of Mesenchymal Stem Cells as a Tool for Proliferation and Differentiation in Cartilage Tissue Engineering: A scaffold-based approach” represents a contribution to field of research in Bioengineering science.

The main problem that the research deals with is the examination of the possibility of applying electrical stimulation in the area of cartilage tissue engineering.

The manuscript fills the gap in the existing published results, proposing a method and methodology that could enable the induction of chondrogenic differentiation and proliferation, which is important for the treatment of osteoarthritis. The deficiency of the manuscript is that it lacks in vivo testing.

Text is relative clear and relative easy to read.

The proposed research concept is original.

The conclusions are in accordance with the presented evidence and arguments.

The literature used is adequate.

Before accepting the manuscript, it is essential that the authors (make corrections):

  1. The goal: “This study aims to utilize ES for promoting CD and proliferation with minimal cell stress”. You need to write two to three sentences, please. One sentence for the goal is unusual.
  2. Figure 3a, b, c and d., it is necessary to mark the magnification, add a scale to the images.
  3. Figure 5a. After 4 days, cell death in the stimulated system is significantly higher than in the not stimulated system. A similar result was obtained after 7 days. Does this result justify the concept you wanted to develop in the submitted manuscript? I would like to ask for a more detailed explanation.
  4. Please state the reason for your choice of SOX9 gene expression in research (Figure 7d).

Round 2

Reviewer 2 Report

Comments and Suggestions for Authors

The author provided satisfactory answers to the questions we raised. Although there have been no additional experiments in some parts, the overall quality has been improved obviously after revision. Therefore, my recommendation is accept.

Author Response

Dear reviewer,

thank you very much for your positive review and your recommendation to accept.

Best regards.